# Towards Generalization in Subitizing with Neuro-Symbolic Loss using Holographic Reduced Representations

**Mohammad Mahmudul Alam** [1], **Edward Raff** [1, 2, 3], **Tim Oates** [1]

[1] University of Maryland, Baltimore County
[2] Booz Allen Hamilton
[3] Syracuse University

## Abstract

While deep learning has enjoyed significant success in computer vision tasks over the past decade, many shortcomings still exist from a Cognitive Science (CogSci) perspective. In particular, the ability to subitize, i.e., quickly and accurately identify the small ($\leq 6$) count of items, is not well learned by current Convolutional Neural Networks (CNNs) or Vision Transformers (ViTs) when using a standard cross-entropy (CE) loss. In this paper, we demonstrate that adapting tools used in CogSci research can improve the subitizing generalization of CNNs and ViTs by developing an alternative loss function using Holographic Reduced Representations (HRRs). We investigate how this neuro-symbolic approach to learning affects the subitizing capability of CNNs and ViTs, and so we focus on specially crafted problems that isolate generalization to specific aspects of subitizing. Via saliency maps and out-of-distribution performance, we are able to empirically observe that the proposed HRR loss improves subitizing generalization though it does not completely solve the problem. In addition, we find that ViTs perform considerably worse compared to CNNs in most respects on subitizing, except on one axis where an HRR-based loss provides improvement. Code is available on GitHub.[1]

## Introduction

Subitizing, also referred to as numerosity, is the ability to recognize small counts nearly instantaneously (Kaufman et al. 1949), allowing for fast, accurate, and confident identification of an object's count in limited space. The ability to recognize drops quickly after four items (Saltzman and Garner 1948). Subitizing is a cognitive function distinct from explicit counting (Trick and Pylyshyn 1994), and recent work has shown that Convolutional Neural networks (CNNs) fail to subitize on simple MNIST-like tasks (Wu, Zhang, and Shu 2019).

The failure is astonishing because a simple, hard-coded convolutional kernel is capable of perfectly solving the subitizing tasks (Wu, Zhang, and Shu 2019). This means a CNN captures the hypothesis space of a valid solution, so it is unclear what component is unable to reach this target goal. Seemingly there are two options: the need for better

Neuro-Symbolic Learning and Reasoning in the Era of Large Language Models (NuCLeaR) Workshop at AAAI 2024.

[1]GitHub: https://github.com/MahmudulAlam/Subitizing

optimization strategies, or alternative loss functions. While a different loss function may sound implausible when using cross-entropy (CE) on a simple, clean dataset, we explore changing the loss function as the strategy in this work.

The goal of this work is to investigate how a neuro-symbolic approach affects the generalization of subitizing in a CNN, but not to solve the problem. We devise a prediction and loss strategy built from the Holographic Reduced Representations (HRRs) (Plate 1995) which has a long successful history of its use in Cognitive Science (CogSci) research.

The proposed loss function is applied to the same set of experiments as proposed by (Wu, Zhang, and Shu 2019) where a CNN failed to subitize. Our results indicate an improvement in generalization on most of the tasks under consideration but are not yet a complete answer to the subitizing task. Favorably, the errors in generalization with our approach are more congruent with the expectation that performance will decrease after 5 objects are present, though the accuracy is still lower than human performance. Moreover, the same set of experiments is performed on a Vision Transformer (ViT) (Dosovitskiy et al. 2020) where the proposed loss function demonstrates improvement in generalization over CE loss and results are more in accordance with subitization expectation as well.

In summary, our contributions are: 1) An adaption of the HRR into a loss function for classification. 2) A empirical evaluation of the impact of subitizing, and a qualitative evaluation of the cases where subitizing is improved or hindered based on the loss function. Note that improved predictive accuracy is not a goal, and difficult to deconflate from subitization performance due to background items. In addition, classic object detection methods (e.g., FasterRCNN (Ren et al. 2015)) are not a proxy of subitizing because such methods perform explicit object counting, where subitizing is a task of instantaneous recognition of numerosity — not a sequential process of identification and counting.

The remainder of the paper is organized as follows. First, different types of vector symbolic architectures, related works, and our motivation for using HRRs are covered. Next, a brief overview of HRRs is provided and the methodology of the proposed HRR loss function is described. Afterward, all the experiments and the corresponding results are described. Finally, concluding remarks, limitations, and future work are presented.

## Related Work

Vector Symbolic Architectures (VSA) have been researched since seminal work by (Smolensky 1990), who made an ever-green argument for their use. In short, VSAs provide a foundation for combining the benefits of connectionist architectures (robustness to deviations in input, and learning) with the benefits of symbolic AI (reasoning, logical inference). This is made possible by defining a system in which arbitrary concepts are assigned to specific vectors, and a set of *binding* and *unbinding* operations are defined, which associate or disassociate two vectors respectively (Schlegel, Neubert, and Protzel 2021). Most VSAs use a fixed feature space for their representation, and thus necessarily introduce noise as more items are bound/unbound. Barring this noise they can symbolically manipulate the concepts associated with the original vectors.

Many such VSAs exist today (Gosmann and Eliasmith 2019; Gayler 1998; Gallant and Okaywe 2013; Kanerva 1996). For example, given vectors representing running, sleeping, cat, and dog, one can compose a vector $\boldsymbol{x} = $ bind(*running*, *cat*) + bind(*sleeping*, *dog*), and then generally determine which animal was sleeping by computing unbind($\boldsymbol{x}$, *sleeping*) $\approx dog$. While the specifics vary between VSAs, we will use the Holographic Reduced Representation proposed by (Plate 1995), which is both commutative and associative in the binding and unbinding operations and has been used successfully in multiple differentiable applications (Alam et al. 2022, 2023; Saul et al. 2023; Menet et al. 2023).

The motivation for using HRR is that it may specifically engender better subitizing which is inspired by current literature in CogSci research that leverages the HRR. The seminal work by (Eliasmith et al. 2012) developed "Spaun,"(Choo 2018) a visual input-based brain model implemented using HRRs and able to perform several cognitive tasks like counting, question answering, rapid variable creation, and others. The HRR has been implemented in a spiking infrastructure (Bekolay et al. 2014) for biological plausibility, but has also shown utility in analogy reasoning (Eliasmith and Thagard 2001), and solving Raven's Progressive Matrices (Rasmussen and Eliasmith 2011).

Little work has been done investigating subitizing via machine learning. Early work by (Zhang et al. 2015) treated the classification task from a purely ML perspective looking for enhanced performance. Later work showed that endowing an object segmentation network with the subitizing task improved the saliency of individual object recognition (He et al. 2017; Islam, Kalash, and Bruce 2018). Our work is concerned with the generalization of subitizing in simple images, which a CNN is not able to do, as shown by (Wu, Zhang, and Shu 2019). We use their MNIST-like shape, color, and edge generalization tasks to measure if an HRR-based loss function can improve the generalization of subitizing in simple CNNs (Wu, Zhang, and Shu 2019). This allows us to isolate the problem to just subitization, and show that the HRR loss does improve results for most generalization tasks.

Due to the severe deficiency of modern CNNs to subitize simple images, we consider many possible related tasks out of scope in our study. This includes prior work in other visual aspects like foveation (Kaplanyan et al. 2019) and visual reasoning (Nie et al. 2020), which intersect machine learning and CogSci. Our goal is only to study how a tool in CogSci modeling, the HRR, impacts CNNs' robustness to the cognitive task of subitizing. Because CNNs cannot yet perform the task at human levels, we also consider matching human reaction times and performance matters for future work.

## Methodology

### Background

Before diving into the construction of our loss function, we will first review the details of the HRR. HRRs are a type of VSA that represent compositional structure using circular convolution in distributed representations (Plate 1995). Given vectors $\boldsymbol{x}_i$ and $\boldsymbol{y}_i$ in a $d$-dimensional space $\mathbb{R}^d$, Plate (1995) used a circular convolution to define a *binding* operation between these two vectors sampled from a Normal distribution. This can be specified more succinctly using the Fourier transform $\mathcal{F}(\cdot)$ and its inverse $\mathcal{F}^{-1}(\cdot)$. Specifically, the resulting vector $\mathcal{B} \in \mathbb{R}^d$ of binding $\boldsymbol{x}_i$ and $\boldsymbol{y}_i$ is given by $\mathcal{B} = \boldsymbol{x}_i \oplus \boldsymbol{y}_i = \mathcal{F}^{-1}(\mathcal{F}(\boldsymbol{x}_i) \odot \mathcal{F}(\boldsymbol{y}_i))$ where $\odot$ indicates element-wise multiplication. Here we use the symbol $\oplus$ to denote the binding operation.

The retrieval of bound components is referred to as *unbinding*. A vector can be retrieved by constructing an inverse function $\dagger : \mathbb{R}^d \rightarrow \mathbb{R}^d$ so that it complies with the identity function $\mathcal{F}(\boldsymbol{z}_i^{\dagger}) \cdot \mathcal{F}(\boldsymbol{z}_i) = \vec{1}$ where $\boldsymbol{z}_i^{\dagger}$ is the inverse of the vector $\boldsymbol{z}$ given by $\boldsymbol{z}_i^{\dagger} = \mathcal{F}^{-1}(1/\mathcal{F}(\boldsymbol{z}_i))$. To unbind $\boldsymbol{x}_i$ from $\mathcal{B}$, we circularly convolve its inverse: $\mathcal{B} \oplus \boldsymbol{x}_i^{\dagger} \approx \boldsymbol{y}_i$. The necessary condition for these operations to behave as expected is an initialization procedure. As originally proposed by (Plate 1995), each vector is sampled from a Normal distribution as $\boldsymbol{z}_i \sim \mathcal{N}(0, 1/d)$. This sampling means that in expectation, the above binding and unbinding steps will work for random pairs of vectors. However, the inversion operation is numerically unstable, and originally a pseudo-inverse was proposed that traded a large numerical error for a smaller approximation error. However, more recently (Ganesan et al. 2021) proposed a projection operation $\pi(\cdot)$ to enforce that the inverse will be numerically stable, and exactly equal to the faster pseudo-inverse of (Plate 1995). This is done by a projection $\pi(\cdot)$ onto the ball of complex unit magnitude, $\pi(\boldsymbol{z}_i) = \mathcal{F}^{-1}\left(\mathcal{F}(\boldsymbol{z}_i)/|\mathcal{F}(\boldsymbol{z}_i)|\right)$. We make use of this projection step to initialize the vectors in our work.

### HRR Loss Function

In this paper, experiments are performed using both CNN and ViT models that take an image as input and predict the number of objects present in that image. To train such models, a standard softmax cross entropy (CE) loss can approximate the one-hot representation of the associated class/count. In our approach, we have taken a different strategy to devise the HRR loss function. We re-interpret the logits of CNN and ViT as an HRR vector instead of approximating a one-hot encoding. We then convert the logits to a class prediction by associating each class with its own unique HRR

vector. To keep the comparison with CE loss fair, our HRR loss will maintain a classification style design in which each class corresponds to a distinct count of objects[2].

The idea here is to represent each class with a unique key-value $(\mathbf{K} - \mathbf{V})$ pair identifier. Each $\mathbf{K}$ and $\mathbf{V}$ is uniquely sampled from normal distribution with projection $\pi(\mathcal{N}(0, \mathbf{I}_H \cdot H^{-1}))$ where $H$ is the feature size. We use the concept of *binding* and *unbinding* operations of HRRs and the network will predict the linked key-value pair, i.e., the bound term. Therefore, if the unbinding operation is performed using the key $k_n \in \mathbf{K} = \{k_1, k_2, \cdots, k_C\}$ where C is the number of classes, the associated value vector $v_n \in \mathbf{V} = \{v_1, v_2, \cdots, v_C\}$ is expected to be the output, $\mathbf{K}, \mathbf{V} \in \mathbb{R}^{1 \times C \times H}$.

Let a network $\mathbf{F}$ predict bound vector $\hat{\mathbf{Y}} \in \mathbb{R}^{B \times 1 \times H}$ of feature size $H$ with $\tanh$ activation function in the final layer for input $\mathbf{X}$ of batch size $B$. The choice of $\tanh$ activation is intentional to keep the output in the range of $[-1, 1]$ as $\mathbf{K} \oplus \mathbf{V}$ will remain in this range. This is due to sampling from a normal distribution with mean zero and standard deviation $1/\sqrt{H}$. $99.98\%$ of the data will be in the following range $-4/\sqrt{H} < k_n, v_n < 4/\sqrt{H}$ ($4\sigma$ rule where $\sigma$ is the standard deviation). Therefore, it is safe to assume that the extremum of $k_n \oplus v_n$ would be $\leq |4\sqrt{2}/\sqrt{H}|$. Choosing a sufficiently large value of $\{H : H \gg 32\}$ would keep the value of $\mathbf{Y} = \mathbf{K} \oplus \mathbf{V}$ in the $[-1, 1]$ range.

To make sure that the network predicts the linked key-value pair associated with the input class of the image, the loss function is defined by Equation 1, where $\hat{y}_i \in \hat{\mathbf{Y}} = \tanh(\mathbf{F}(\cdot))$ is the network's output.

Equation 1 is sufficient for training the network, but we still need an explicit prediction for evaluation. To get the associated class label from the network output, we apply the $\mathbf{K}$ vectors of all the C classes to the $\hat{\mathbf{Y}}$ which will return the estimation of value vectors $\hat{\mathbf{V}} = \mathbf{K} \oplus \hat{\mathbf{Y}} \in \mathbb{R}^{B \times C \times H}$. $\hat{\mathbf{V}}$ contains the values for all the C classes, however, the value for the associated input would be the most similar to the ground truth value after training. Accordingly, the cosine similarity score $\mathbf{S}$ is calculated given in Equation 2, and the $\arg\max$ of $\mathbf{S}$ will be the predicted class/count output associated with the input image.

$$\mathcal{L} = \sum_{i=1}^{B} \| k_i \oplus v_i - \hat{y}_i \|_2 \tag{1}$$

$$\mathbf{S} = \frac{\sum_{i=1}^{H} \mathbf{V}_i \cdot \hat{\mathbf{V}}_i}{\|\mathbf{V}\|_2 \|\hat{\mathbf{V}}\|_2} \in \mathbb{R}^{B \times C} \tag{2}$$

## Experiments and Results

Wu, Zhang, and Shu (2019) examined the cognitive potential of a CNN in numerosity using four experiments. Numerosity is perhaps the simplest innate cognitive computing task

that a child can do. Disappointingly, the key finding of the work is the failure in the subitizing tasks of the CNN learned by CE loss. In this paper, we re-do the same experiments using the same CNN to show how our proposed HRR loss function, where each class is represented using a unique key-value pair, improves the CNN's numerosity performance.

Humans have a good sense of small numbers and can recognize the number of objects in a scene up to 4 items without counting them explicitly (Nieder and Miller 2003; Piazza et al. 2004; Tokita and Ishiguchi 2010). This ability is independent of the type, shape, and color of the object. For example, if a child learns to subitize or count circles, that same skill is utilized to subitize or count squares even though circles and squares have different shapes. Nevertheless, current methods of training CNNs on subitizing perform poorly in comparison to humans.

In the following experiments, we discuss how the basic skills of numerosity are lacking in CNNs and how the proposed loss helps to build a numerical sense. In all these experiments, the same CNN and dataset are used as in (Wu, Zhang, and Shu 2019). In addition, a ViT network is used in the same set of experiments. However, we modify the final layer of the networks with the HRR loss. Instead of predicting logits with softmax activation from the network, the network is used to predict features of size $H = 64$ with a $\tanh$ activation function for both networks.

The network is trained using the Numerosity database which has a total of 6000 training images of dimension $100 \times 100$ with a varying number of circles from 1 to 6. The test dataset contains 7 variations (described below) of the training images. Each variation of the test split contains 6000 images [3].

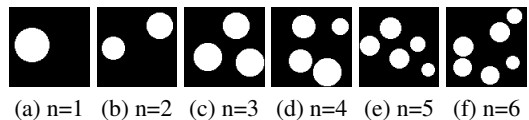

(a) n=1  (b) n=2  (c) n=3  (d) n=4  (e) n=5  (f) n=6

Figure 1: Sample training images of classes 1 to 6 are shown from (a) to (f) used to train the network for the first four experiments. The task is to predict the number of objects in an image. The generalization is tested using five different test sets in four groups that alter the size, shape, color, and infilling of the objects to make the task more difficult.

The training set contains images of white circles on a black background. They are made such that the number of circles is independent of the total area of the circles to avoid any possible information leakage that may be used to "cheat" and obtain predictions without learning to actually subitize. The maximum number of circles, i.e., the total number of classes, is $C = 6$. A sample image of each class is given in Figure 1. For ViTs, images are divided into $10 \times 10$ patches. For each patch, a feature of size 256 is used. In multi-head attention, 4 heads are used and the encoder block is repeated 6 times. Both networks are trained

[2]One could select the HRR vectors to encode an ordinal style loss, but that amounts a prior for counting in the loss design. Our goal is to determine if the HRR alone has benefits separate from being able to implement inductive biases into the architecture. Thus a classification-oriented design maintains that goal.

[3]Training and test images are not publicly available. We got access to the dataset in correspondence with (Wu, Zhang, and Shu 2019).

by optimizing the HRR loss function in Equation 1 for a total of 300 epochs on a single RTX 2070 Super 8GB GPU. The dropout rate is set to be $0.1$ and the initial learning rate is set to be $10^{-3}$ for the first 100 epochs which is lowered to $10^{-4}$ and $10^{-5}$ for every 100 epochs.

Framing the task in terms of classification presents challenges when interpreting the results. There are cases where the network consistently over-predicts the true number of items in an image (i.e., says "4" instead of "3"). This causes cases of false success, in that the accuracy of predicting the target of "6" is near 100% not because the network has successfully subitized, but because the network cannot over-predict beyond 6, and through this limit falsely appears to perform well. This situation is common, and *we identify such cases with italics* to avoid incorrectly bringing the reader's attention to what is actually a failure, while simultaneously indicating the nature of the result. This also occurs with consistent under-counting and the "1" target class but is less prevalent in the results.

With this caveat, we describe the set of experiments that were performed and their results. In the following subsections, the subitizing ability of a CNN and ViT is tested and compared using both CE and HRR loss. We also show saliency maps (Simonyan, Vedaldi, and Zisserman 2013) for each example test image. The saliency maps allow us to better understand why the HRR approach improves subitizing in the majority of cases over CE loss. The general result is that the standard cross-entropy loss has spurious attention placed on non-informative regions of the image. The HRR approach is not immune to this, especially since the network between approaches is the same, but it is noteworthy how significant the difference is.

## Experiment of Object Sizes

The networks are originally trained using the images of circles shown in Figure 1 and it classifies all the training images with 100% accuracy. In this experiment, we test the performance of the network with the test images of circles where the size of the circles is made 50% larger than the original training images. Apart from that, all other parameters such as color and shape are kept the same. The sample images of the circle with a bigger radius are illustrated in Figure 2. Results of this experiment are presented in the '50% Larger' column of Table 1 and Table 2 for the CNN and ViT, respectively. Although varying object size does not cause the CE network's accuracy to fall significantly for classes 1 to 4, for classes 5 and 6 of the CNN, and for class 5 of the ViT, accuracy falls considerably. On the other hand, HRR loss can classify all the images with over 80% accuracy using the CNN and over 50% accuracy using the ViT for all the classes. It is interesting to note that the accuracy follows the subitizing pattern, i.e., as the number of circles in the image increases the probability of correctly recognizing them decreases. Figure 2 shows the saliency maps of both HRR and CE loss for the CNN. HRR loss puts more restricted attention in the boundary regions whereas attention in the case of the CE loss spreads out broadly.

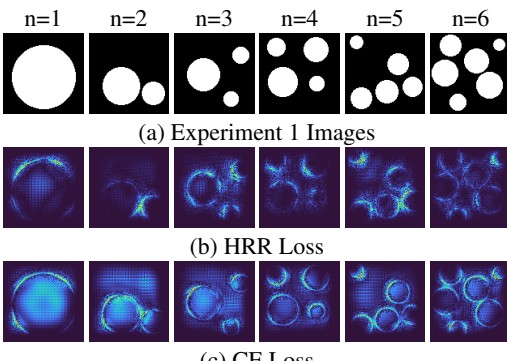

(a) Experiment 1 Images

(b) HRR Loss

(c) CE Loss

Figure 2: Sample images of experiment 1 where the radius of the circles are $50\%$ greater than the circles of training images are shown in (a). Saliency maps of the experiment 1 images for both HRR and CE loss are shown in (b) and (c), respectively. HRR puts more attention toward the boundary regions whereas the network trained with CE loss function puts attention on both the inside and output of circles along with the boundary regions.

## Experiment of Object Shapes

In this experiment, the networks are tested by replacing the circles with other shapes such as white equilateral triangles and squares on a black background, illustrated in Figure 3. Results of this experiment are presented in the 'Triangles' and 'Squares' columns of Table 1 and Table 2. When only changing the shape of the object to triangles, the accuracy of the CE CNN drops below $50\%$ for all classes except for class 6, with an average accuracy of $45.17\%$, revealing poor generalization. In the case of the images of squares, the network performs comparably well with an increase in average accuracy to $75.68\%$. By contrast, due to using the HRR loss and a key-value-based transformation layer, the accuracy of the same network is over $50\%$ for images of triangles and over $80\%$ for images of squares for all the classes. The average accuracy for triangles and squares is $75.7\%$ and $77.0\%$, respectively. In the case of ViT, the performance of both HRR and CE losses are similar. For images of triangles, the HRR loss average accuracy is $55.33\%$, slightly lagging behind the CE loss accuracy of $56.0\%$, whereas for images of squares, the HRR loss average accuracy is $65.66\%$, slightly lagging behind the CE loss accuracy of $66.0\%$. The saliency maps for both HRR and CE loss for the CNN are presented in Figure 3. Consistently, the HRR loss puts strict focus on the edges of the objects whereas the CE loss spreads attention throughout the image.

## Experiment of Object Colors

The object's color in the test images is swapped in this experiment. The images contain newly generated synthetic circles of the same size as the training set circles, but the test circles are black on a white background. The results of this experiment are the 'Color Swap' column of the Table 1 and Table 2. Figure 4 shows the example images that are used in this experiment along with the saliency maps. From the

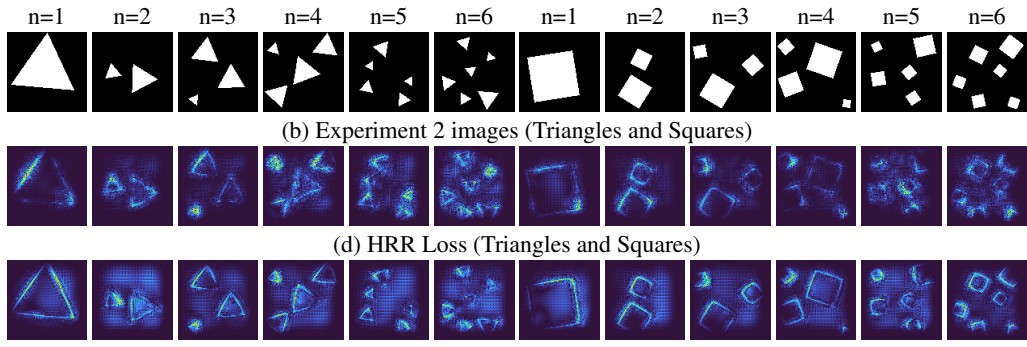

|  | n=1 | n=2 | n=3 | n=4 | n=5 | n=6 | n=1 | n=2 | n=3 | n=4 | n=5 | n=6 |

(b) Experiment 2 images (Triangles and Squares)

(d) HRR Loss (Triangles and Squares)

(f) CE Loss (Triangles and Squares)

Figure 3: Sample images of experiment 2 where circles of classes 1 to 6 are replaced by triangles and squares shown in (a). Filters that rely on the curvature of a circle explicitly will perform poorly on this task, which is evident in the CE approach's lower accuracy. Saliency maps of the experiment 2 images are shown in (b) for HRR loss and (c) for CE loss. HRR's attention is concentrated on the informative regions, i.e., boundary regions whereas attention is more distributive in the case of CE.

| | 50% Larger | | Triangles | | Squares | | Color Swap | | White Rings | |
|---|---|---|---|---|---|---|---|---|---|---|
| Target | HRR | CE | HRR | CE | HRR | CE | HRR | CE | HRR | CE |
| 1 | 1.000 | 1.000 | **0.997** | 0.327 | **1.000** | 0.876 | 0.093 | 0.160 | 0.033 | 0.004 |
| 2 | 0.920 | **0.997** | **0.787** | 0.441 | **0.914** | 0.811 | 0.228 | **0.340** | 0.007 | 0.002 |
| 3 | 0.967 | **0.990** | **0.715** | 0.361 | **0.964** | 0.641 | 0.388 | **0.680** | 0.000 | 0.010 |
| 4 | 0.953 | **0.959** | **0.541** | 0.287 | **0.944** | 0.686 | 0.370 | **0.670** | 0.003 | 0.096 |
| 5 | **0.904** | 0.672 | **0.619** | 0.364 | **0.900** | 0.549 | 0.251 | **0.420** | 0.019 | **0.194** |
| 6 | **0.815** | 0.549 | 0.883 | *0.930* | 0.888 | *0.978* | 0.122 | **0.250** | *1.000* | *0.989* |

Table 1: Results of the CNN where **bold** are best unless the result is due to consistent *over/under accounting at the boundary*. No result is marked "best" when performance is worse than random guessing ($\leq 16.7\%$) or similar. The HRR approach generalizes better for the first three tasks (or is closely behind) but degrades on the color swap task. Both methods fail on the last test.

| | 50% Larger | | Triangles | | Squares | | Color Swap | | White Rings | |
|---|---|---|---|---|---|---|---|---|---|---|
| Target | HRR | CE | HRR | CE | HRR | CE | HRR | CE | HRR | CE |
| 1 | 1.000 | 1.000 | 0.637 | **0.681** | 0.942 | **0.977** | **0.020** | 0.000 | **0.632** | 0.053 |
| 2 | 0.932 | **0.981** | 0.595 | **0.662** | 0.731 | **0.798** | **0.026** | 0.001 | **0.616** | 0.113 |
| 3 | 0.920 | 0.923 | **0.488** | 0.470 | 0.553 | **0.567** | **0.062** | 0.001 | **0.467** | 0.187 |
| 4 | 0.780 | 0.785 | **0.356** | 0.331 | **0.393** | 0.340 | **0.094** | 0.005 | **0.366** | 0.331 |
| 5 | **0.555** | 0.372 | **0.431** | 0.312 | **0.401** | 0.276 | **0.283** | 0.024 | 0.267 | **0.382** |
| 6 | 0.990 | 0.995 | 0.813 | **0.906** | 0.948 | **0.968** | 0.822 | *0.995* | 0.269 | *0.704* |

Table 2: Results of the ViT where **bold** are best unless the result is due to consistent *over/under accounting at the boundary*. No result is marked "best" when the performance of both methods is comparable. The HRR approach generalizes better or closely behind for all the tasks while using ViT. In the color swap task, we can see performance degrades for both but HRR yields better generalization.

figure, it is obvious that the changes in the test images are immense compared to the training images from a network's perspective. From a human perspective, this is quite an easy task to generalize after learning from the training images. Both of the methods also fail the subitizing test. A human being can count a lower number of objects with less effort than a higher number of objects. Nevertheless, the CE classification approach has achieved 16% accuracy for class 1 and 25% for class 6. Likewise, the HRR-based method has achieved 9.3% for class 1 and 12.2% for class 6. However,

in the case of the ViT, while the performance using both losses degrades and degenerates, the HRR loss shows better generalization compared to the CE approach.

**Experiment of Region-Boundary Duality**

Differentiating between objects from the boundary representation is vital to recognition (Marr 2010). Humans can easily identify objects, separate and count objects given just their boundaries. To examine the network's ability to generalize across the region-boundary duality, the network is

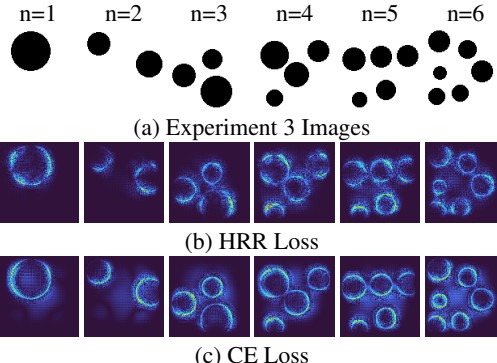

(a) Experiment 3 Images

(b) HRR Loss

(c) CE Loss

Figure 4: Sample images of experiment 3 where the circle and background colors are swapped in the test images shown in (a). Saliency maps of the HRR and CE loss are shown in (b) and (c), respectively. The attention of the network is more focused on the boundary region in the case of HRR.

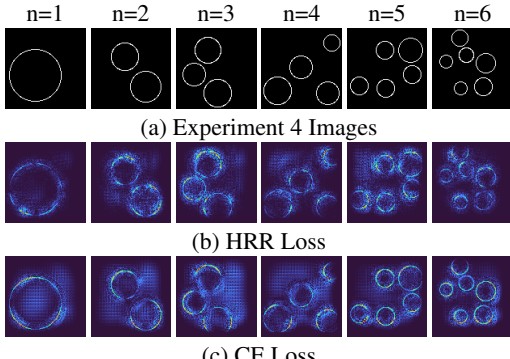

(a) Experiment 4 Images

(b) HRR Loss

(c) CE Loss

Figure 5: Sample images of experiment 4 where the circles are represented by the boundary edges shown in (a). This is the most challenging generalization task, as it changes the ratio of white and black pixels. Saliency maps for object region-boundary duality are shown in (b) and (c) for HRR and CE, respectively.

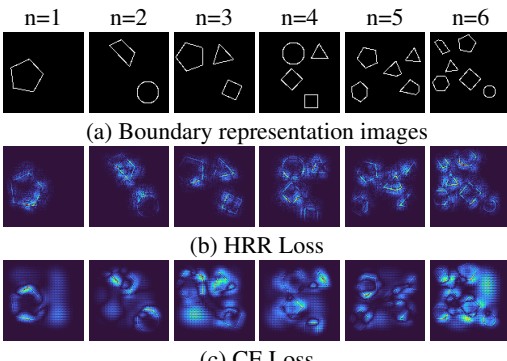

(a) Boundary representation images

(b) HRR Loss

(c) CE Loss

Figure 6: Sample images of boundary representation of the various shaped objects are shown in (a). In all cases with the CE loss shown in (c), we see spurious attention placed on empty regions of the input - generally increasing in magnitude with more items. By contrast, the HRR loss shown in (b) keeps activations focused on the actual object edges and appears to suffer only for large $n$ when objects are placed too close together.

tested using images of white circle rings on a black background. Examples of these test images along with saliency maps are presented in Figure 5, and the results are in the 'White Rings' columns of Table 1 and Table 2.

Recall that the network is originally trained on the images in Figure 1. From the network's perspective, the rings of white circles are completely new images. As a result, both the CE classification approach with softmax activation and the HRR classification approach with the key-value transformation layer approach degrade in performance. In the case of CNN, we can see degeneracy for both CE and HRR losses except for class 6 where both methods overcount and have achieved $98.9\%$ and $100\%$ accuracy, respectively. This is peculiar from the subitizing point of view because the accuracy for classes with a single ring of a circle in each approach is $0.4\%$ and $3.3\%$, respectively. However, in the case of the ViT, we can see the effectiveness of the HRR loss over CE loss for classes 1 to 4 with a big margin ranging from $4\%$ to $58\%$. For classes 5 and 6, HRR loss remains consistent with the subitizing pattern with lower accuracy than CE loss, but for class 6 the CE loss overcounts. In conclusion, the CNN lacks the ability to generalize across the region-boundary duality and fails on this more complex subitizing task. On the other hand, the ViT with HRR loss shows robust performance in generalization on this complex subitizing task.

## Boundary Representation Tests

Experiments 1 to 4 demonstrate CNN's lack of generalization in learning. To improve the abstraction ability of CNNs, Wu et. al. (Wu, Zhang, and Shu 2019) suggested learning from the boundary representation of objects. Instead of learning from single-shaped images, each class is built with different-shaped polygons with n sides. This should eliminate the shape bias in test results. The size will be altered to allow isolation of generalization to fundamental subitizing ability rather than change the re-use of shape patterns. Moreover, each object is represented by its boundary which bridges the representation of the black object on a white background and the white object on a black background. Figure 6 illustrates sample images of different shapes and sizes of objects with the boundary representation.

The network is re-trained using $80\%$ of the images of Figure 6 and the remaining $20\%$ of the images is used for testing. The accuracy on a test set of in-distribution is shown in Table 3. While the CE loss appears to obtain better training accuracy, the goal of this study is the generalization of subitizing ability. As such the results in Table 3 are more interesting because the in-distribution results are seen to imply that the HRR loss is worse, but we will see that it has a meaningful impact on generalization. This nuance would be difficult to identify in standard computer vision datasets.

To inspect how much generalization is achieved by training the network with images of object boundaries, the test

|        | Boundary Edge Representation | |
|--------|-------|-------|
| Target | HRR   | CE    |
| 1 | 1.000 | 1.000 |
| 2 | 0.985 | 1.000 |
| 3 | 0.950 | 0.970 |
| 4 | 0.855 | 0.930 |
| 5 | 0.635 | 0.790 |
| 6 | 0.795 | 0.920 |

Table 3: In distribution results, show baseline training performance of the HRR and CE-based loss functions on the edge-map distribution, rather than testing generalization. In practice, while the HRR has a lower training accuracy, it has better generalization.

images are scaled up and down by $50\%$. Next, we will examine how boundary representation helps towards generalization. Intriguingly, the CE method does not follow the expected subitizing degradation pattern, though our HRR approach is closer to achieving it for the 50% larger case.

Table 4 reveals how the results deteriorate by only changing the scale of the object. However, in the case of scaling up, both of the methods show solid evidence of human-like subitizing, i.e., the accuracy decreases as the number of objects in the image increases. The proposed HRR loss approach has achieved an average accuracy of $49\%$ whereas the CE approach has achieved an average accuracy of $45.6\%$, but the CE's performance is inflated in the sense that it has a higher training accuracy and drops precipitously.

|        | 50% Larger | | 50% Smaller | |
|--------|-------|-------|-------|-------|
| Target | HRR   | CE    | HRR   | CE    |
| 1 | 0.935 | **0.991** | *1.000* | 0.687 |
| 2 | 0.715 | **0.984** | 0.005 | **0.390** |
| 3 | **0.585** | 0.496 | 0.005 | 0.021 |
| 4 | **0.300** | 0.207 | 0.000 | 0.014 |
| 5 | **0.225** | 0.032 | 0.000 | 0.043 |
| 6 | **0.180** | 0.026 | 0.000 | *0.988* |

Table 4: Generalization results for the boundary edge maps. **Bold** results are the best unless the result is due to *over/under accounting at the boundary*. No result is marked "best" when worse than random guessing ($\leq 16.7\%$).

In the case of scaling down, no apparent subitizing pattern is present for either method. The proposed method achieved $100\%$ accuracy for class 1 due to under-counting and failed to generalize for the rest of the classes. Conversely, the CE approach has achieved $98.8\%$ accuracy due to over-counting for class 6 and failed to generalize for the rest of the classes. Overall, the boundary representation has helped the network's abstraction ability of subitizing but failed to generalize, especially in the case of scaling down.

The saliency maps of the boundary representation test images are presented in Figure 6. In the boundary representation tests, decisions are supposed to be made by the edge/boundary representation. The saliency maps reveal how HRR loss is concentrating networks' attention in the boundary regions whereas attention is much diffused in the case of CE loss. Moreover, based on the observation of saliency maps of correct and incorrect predictions following conclusions (see Appendix A for details) are made:

- Even when the CE-based model is **correct**, its saliency map indicates it uses the inside region of an object and the area around the object/background toward its prediction in almost all cases.
- When the HRR loss-based model is **correct**, it rarely activates for anything besides the object boundary and does not tend to focus on the inside content of an object.
- When the HRR-based model is **correct**, the edges of the objects in the saliency map are usually nearly-complete, and large noisy activations can be observed surrounding the boundary regions.
- When the CE-based model is **incorrect**, it often has two objects that are nearby each other. When this happens, the CE saliency map tends to produce especially large activations between the objects, creating an artificial "bridge" between the two objects.
- When the HRR-based loss is **incorrect**, it tends to have a saliency map that is either 1) activating on the inside content of the object, or 2) has large broken/incomplete edges detected for the object.

## Conclusion and Future Work

In this paper, a neuro-symbolic loss function is proposed using HRR to investigate the subitizing ability of deep learning networks such as CNN and ViT. In the four experiments, the HRR-based loss appears to improve the results, especially toward higher subitizing generalization. ViT performed comparatively worse than CNN, however, in general, ViT with HRR loss shows better generalization. In one case of CNN, HRR's performance has degraded, but still non-trivial performance, and in one case both the HRR loss and CE loss have degenerated worse-than-random guessing. In the case of ViT, HRR's effectiveness in generalization remains consistent particularly in 'white rings' where it outperformed CE over a big margin ranging from $4\%$ to $58\%$.

Our results are intriguing in that we did not design the HRR loss to be biased toward numerosity via symbolic manipulation, but instead defined a simple loss function as a counterpart to the CE loss that retains a classification focus. This may imply some unique benefit to the HRR operator in improving generalization and supports the years of prior work using it for CogSci research.

While more work remains to improve innate subitizing generalization, we are not yet ready to move past these simplistic benchmarks. While (Wu, Zhang, and Shu 2019) have thoroughly accounted for many potential information leakage sources, the under and over-counting bias remains a limitation to our work and others. This need for improved experimental design of simple tasks also highlights the general need to thoroughly test CNN and ViT broadly and the limitations and likelihood of encountering out-of-distribution data.

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

# Saliency Maps Reviews

The saliency maps of the correct and incorrect predictions by the network both in the case of CE and HRR loss are observed. Example images along with saliency maps for CE loss are given in Figure 7 for correct prediction and in Figure 8 for incorrect predictions. When a network trained with CE loss makes a correct prediction, its saliency maps show it uses the inside region of an object and the area around the object/background toward its prediction in almost all cases.

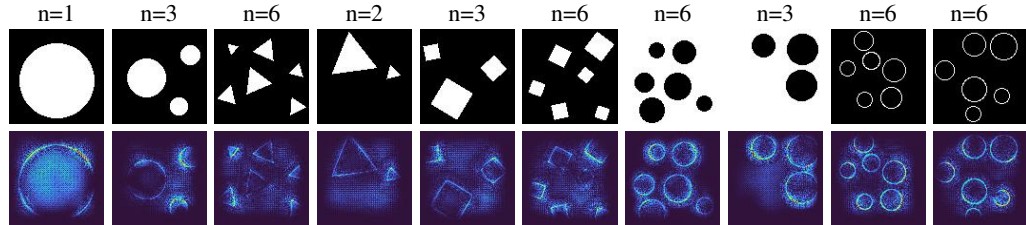

Figure 7: Sample images with saliency maps in a CE-based model for **correct** predictions.

However, when a CE-based model makes an incorrect prediction, often its saliency map tends to produce large activations between the multiple objects, creating an artificial "bridge" among them.

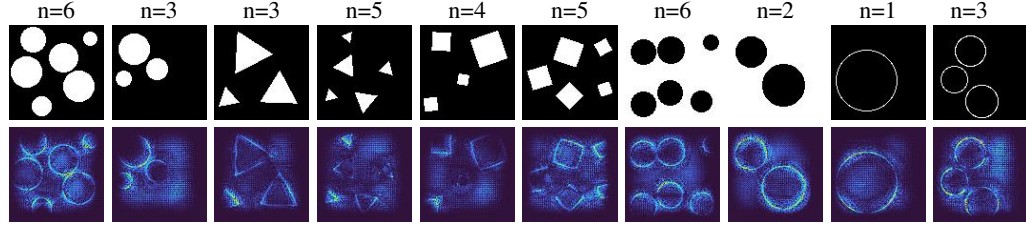

Figure 8: Sample images with saliency maps in a CE-based model for **incorrect** predictions.

Saliency maps along with sample images for HRR-based loss are given in Figure 9 for correct predictions and in Figure 10 for incorrect predictions. While making correct predictions, the edges of the objects in the saliency map of the HRR-based model are usually nearly-complete and we can observe large noisy activations surrounding the boundary regions.

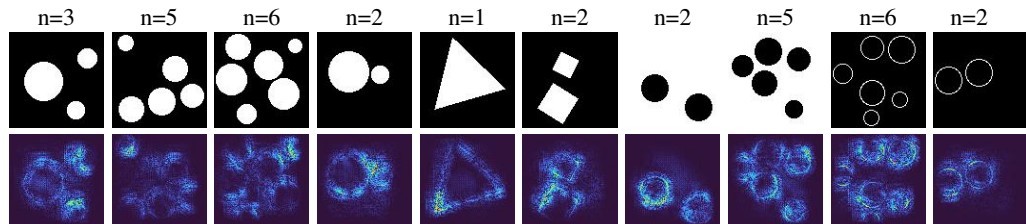

Figure 9: Sample images with saliency maps in a HRR-based model for **correct** predictions.

Nevertheless, when the HRR-based model makes an incorrect prediction, it tends to have a saliency map that is either 1) activating on the inside content of the object, or 2) has large broken/incomplete edges detected for the object.

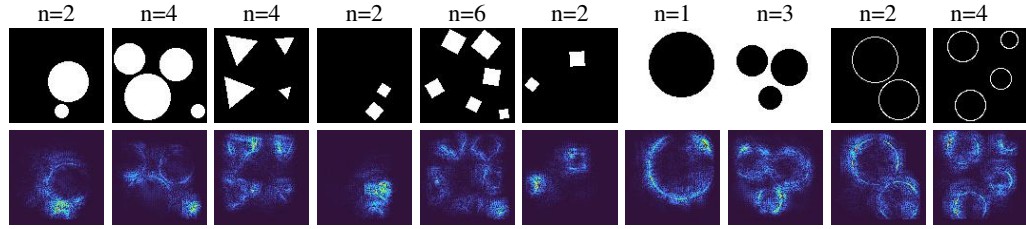

Figure 10: Sample images with saliency maps in a HRR-based model for **incorrect** predictions.