# OpenReview forum: "Towards Generalization in Subitizing with Neuro-Symbolic Loss using Holographic Reduced Representations"
_AAAI.org/2024/Workshop/NuCLeaR — NuCLeaR 2024_

### Official Review · Reviewer_Dckp · 2023-12-06
**Interesting paper with intriguing results**

**Rating:** 8
**Confidence:** 5

**Review:**

This paper makes a empirically-based contribution to the understanding of how CNNs subitize given two different loss functions, a classical CE and a neuro-symbolic HRR. While the paper is well written and the authors made many relevant experiments, the results are a bit weak. However, given the novelty of their approach, their contribution is to be considered as valuable.
Some minor remarks:
* While mentioned in the text, it should be clear in the caption of the tables that the authors use accuracy as their main metric. In the current version of the paper, the authors merely speak of "results".
* It is clearly stated that VSA are regarded as a neuro-symbolic approach and the presentation in this regard is clear. However, it is unclear to me how saliency maps can be used as an appropriate demonstration of neuro-symbolic capabilties. The authors should make clear how saliency maps can reveal symbolic aspects of the inference process in the CNN.

---

### Official Review · Reviewer_UpXW · 2023-12-07
**This paper proposes a new loss function for subitizing using Cog_sci inspired holographic reduced representations, however the practical impact of such an approach is unclear. A detailed analysis comparing object detection methods with the proposed methods is needed.**

**Rating:** 6
**Confidence:** 3

**Review:**

The paper explores an interesting application of neuro-symbolic methods from Cog-Sci as loss functions to enhance the subitizing capability of CNNs. The authors test generalization of their approach across transformations showing minor improvements in limited cases over cross entropy loss. The authors do not provide analysis quantifying the computational efficiency benefits of subitization versus traditional object detection and counting. An analysis of inference speed, latency, or computational cost is needed to demonstrate that the proposed HRR approach retains this advantage over counting detected objects. Without such analysis, the practical speedup of subitizing for real-time applications is unclear.

---

### Official Review · Reviewer_6Asi · 2023-12-08
**The review discusses the proposed approach to enhance subitizing generalization in neural networks through the integration of Holographic Reduced Representations. It highlights the novelty of this integration, its impact on generalization, and suggests future research directions. The review also acknowledges the limitation of using a specialized dataset and recommends exploring broader benchmarks.**

**Rating:** 8
**Confidence:** 4

**Review:**

The authors proposed an innovative approach to enhance the subitizing generalization of Convolutional Neural Networks (CNNs) and Vision Transformers (ViTs). They considered adapting Holographic Reduced Representations (HRRs) to develop an alternative loss function that improves the subitizing capabilities of these neural networks.

I found the integration of HRRs into deep learning loss functions to be novel, bridging the gap between cognitive science and neural network research. The paper also demonstrates that the proposed HRR loss improves generalization in subitizing tasks, particularly for CNNs. The experiment section is extensive, and the saliency map analyses provide empirical support for the effectiveness of the approach. The experiment section also includes a discussion of the comparison between CNNs and ViTs under various conditions, which provides a comprehensive understanding of the models' capabilities. While the authors have clearly mentioned that the HRR loss shows improvement, it does not completely solve the problem of subitizing generalization. Nevertheless, the authors' discussion of the approach's capacity in offering valuable insights and directions for future research in combining symbolic and neural approaches is fair.

The experiments are conducted on a specialized dataset (Numerosity database), which may limit the generalizability of the findings to other datasets or real-world scenarios. I suggest that the authors, as their next step, consider incorporating more benchmarks and datasets to improve the formulation of the approach's generalization.

In summary, the paper presents a significant contribution to the field by introducing a neuro-symbolic approach to enhance the generalization ability of neural networks in cognitive tasks, and I believe it is qualified to be accepted and presented at the NuCLeaR workshop.

---

### Decision · Program_Chairs · 2023-12-11

Accept